# Fast Eating Speed Could Be Associated with HbA1c and Salt Intake Even after Adjusting for Oral Health Status: A Cross-Sectional Study

**DOI:** 10.3390/healthcare11050646

**Published:** 2023-02-23

**Authors:** Satsuki Watanabe, Yuhei Matsuda, Yui Nanba, Mayu Takeda, Takafumi Abe, Kazumichi Tominaga, Minoru Isomura, Takahiro Kanno

**Affiliations:** 1Department of Oral and Maxillofacial Surgery, Shimane University Faculty of Medicine, Izumo 693-8501, Japan; 2Department of Lifetime Oral Health Care Sciences, Graduate School of Medical and Dental Sciences, Tokyo Medical and Dental University, Tokyo 113-8510, Japan; 3Center for Community-Based Healthcare Research and Education (CoHRE), Head Office for Research and Academic Information, Shimane University, Izumo 693-8501, Japan; 4Tominaga Dental Office, Ohchi 696-0313, Japan

**Keywords:** eating speed, oral health status, HbA1c, type 2 diabetes, salt intake

## Abstract

This study aimed to examine the relationship between eating speed and hemoglobin A1c (HbA1c), considering the number of teeth, using cross-sectional health examination data from community-dwelling older individuals in Japan. We used data from the Center for Community-Based Healthcare Research and Education Study in 2019. We collected data on gender, age, body mass index, blood test results, Salt intake, bone mineral density, body fat percentage, muscle mass, basal metabolic rate, number of teeth, and lifestyle information. Eating speed was evaluated subjectively as fast, normal, or slow. Overall, 702 participants were enrolled in the study and 481 participants were analyzed. Multivariate logistic regression analysis revealed a significant association between fast eating speed and being a male (odds ratio [95% confidence interval]: 2.15 [1.02–4.53]), HbA1c (1.60 [1.17–2.19]), salt intake (1.11 [1.01–1.22]), muscle mass (1.05 [1.00–1.09]), and enough sleep (1.60 [1.03–2.50]). Fast eating may be associated with overall health and lifestyle. The characteristics of fast eaters, after taking oral information into consideration, tended to increase the risk of type 2 diabetes, renal dysfunction, and hypertension. Dental professionals should provide dietary and lifestyle guidance to fast eaters.

## 1. Introduction

In a large cohort study in Japan, unhealthy dietary habits were reported to affect obesity and other health conditions [1]. Examples of unhealthy eating habits are snack food consumption, late-night meal consumption, and fast eating [2,3]. Of these, fast eating has been noted to have the potential for widespread health effects [4]. In general, fast eating has been reported to be associated with systemic diseases such as obesity, type 2 diabetes, non-alcoholic fatty liver disease, and renal dysfunction [5,6,7]. Although there is no fixed definition of eating speed, in many studies, a patient’s subjective eating speed is important in the assessment of time [8]. The assessment of self-reported eating speed is also included in the self-administered diet history questionnaire (DHQ), which was developed to evaluate the dietary habits of the Japanese population and has been suggested to be related to eating habits and obesity levels [9]. A meta-analysis has reported a higher body mass index (BMI) when eating faster [8]. Eating speed has also been reported to be significantly associated with a higher risk of metabolic syndrome, elevated blood pressure, and obesity [10]. In addition, eating speed may be associated with proinflammatory cytokines (IL-1β) in Japanese men without metabolic diseases [11]. Conversely, slower eating speeds have been reported to reduce excess food and energy intake [12]. Therefore, eating speed, which is a lifestyle habit, is highly likely to have an impact on health. Focusing on type 2 diabetes, which is strongly associated with metabolic syndrome, it has been reported that fast eating speed is associated with a rapid increase in blood glucose levels [13,14]. Eating speed has also been implicated as an intermediate factor in obesity, and may be associated with diabetes [15]. In addition, some reports suggest that fast eating speeds double the increased risk of type 2 diabetes [16]. However, while gender, age, and BMI have been adjusted for as confounding factors in many studies, few studies have considered oral and dental health status.

Whether oral and dental health status influences eating speed is controversial. The association between masticatory function and eating speed is contradicted by reports that people with fewer dental prostheses eat faster, while a study of 30,938 Japanese adults reported that masticatory difficulty was associated with higher hemoglobin A1c (HbA1c) [17]. Some studies reported that increasing the number of times of mastication decreased the speed of eating, whereas others reported that there was no relationship [18,19]. Next, regarding the relationship between the number of teeth and eating speed, a report found that the probability of having metabolic syndrome was 2.5 times higher in those with a small number of remaining teeth and fast eating than in those with a large number of remaining teeth and slow eating [20]. Additionally, a higher number of remaining teeth and slower eating speed have been reported to reduce the likelihood of metabolic syndrome in the older population [20].

Overall, previous studies suggest that eating speed and oral and dental health status may be related, but few studies have considered such oral and dental health-related factors in studies of eating speed [20,21].

Therefore, two hypotheses were formulated: first, eating speed is associated with oral and dental health-related status, and second, even after adjusting for oral and dental health-related status, systemic health conditions such as type 2 diabetes together with lifestyle conditions and eating speed are associated [20]. This study aimed to examine the relationship between eating speed and systemic health conditions, considering oral and dental health status, using cross-sectional health examination data from community-dwelling older individuals in Japan.

## 2. Materials and Methods

### 2.1. Data Collection

This study used the same dataset as other reports because it is based on datasets obtained from health examinations [22,23]. However, the variables of interest for the analysis and the analysis methods were different. This study was approved by the Medical Research Ethics Committee of Shimane University Faculty of Medicine (number: 20220622-1). Written informed consent was obtained from all participants, and data were collected.

### 2.2. Center for Community-Based Healthcare Research and Education (CoHRE) Study

The CoHRE Study is a cohort study conducted by the Shimane University Center for Community-based Healthcare Research and Education to predict and prevent lifestyle-related diseases in Ohnan-cho, Shimane Prefecture [22,24]. Surveys on health and medical information, various clinical examination information, lifestyle information, human relationship information, social resource information, and medical care cost information are ongoing.

### 2.3. Study Design

This study used cross-sectional data from the 2019 Shimane CoHRE Study; the 2019 data are the most recent version of the dataset because surveys have not been conducted after 2019 due to the COVID-19 pandemic [22].

### 2.4. Inclusion Criteria

The inclusion criteria were as follows: residents covered by Japan National Health Insurance; residents of Ohnan-cho, a mid-mountainous area in Shimane Prefecture, Japan; and residents who participated in the 2019 survey [24].

### 2.5. Exclusion Criteria

Data from residents with missing values were excluded, and complete data were analyzed [22,23].

### 2.6. Collected Data

#### 2.6.1. Background Data

In the CoHRE Study, data were collected annually through standardized questionnaires and physical measurements, blood tests, and urine analysis [22]. We collected data on the following variables: gender (male/female), age, body mass index, high-density lipoprotein (HDL) cholesterol, low-density lipoprotein (LDL) cholesterol, triglyceride, γ-glutamyl transpeptidase, glycemic index, HbA1c, estimated glomerular filtration rate, creatinine, sodium, potassium, salt intake, bone mineral density, body fat percentage, muscle mass, basal metabolic rate, number of teeth, smoking status, physical activity, walking speed (yes/no), sleeping status, and alcohol consumption (every day, sometimes, none). Salt intake was measured by estimating salt intake from spot urine specimens (Tanaka method) [25]. Walking speed and sleep status were binarized as “yes/no”; “yes” corresponded to a walking speed faster than that of an average individual of the same gender at about the same age and when the respondents were well rested, respectively [26].

#### 2.6.2. Eating Speed

As in previous studies, a self-reported subjective assessment method of food intake speed was applied [10,27]. Eating speed was evaluated subjectively on a three-point scale: fast, normal, and slow.

For the analysis, eating speed was treated by dividing the group into two groups: fast and normal/slow.

### 2.7. Statistical Analysis

After confirming the normality of participant data using the Shapiro–Wilk test, continuous data were expressed as means and standard deviations, while categorical data were expressed as numbers (%).

Logistic regression analysis (backward stepwise) was used to control for possible confounding variables related to eating speed. Partial regression coefficients for each eating speed outcome were estimated after adjusting for all other variables included in the model. Adjustment items included gender, age, body mass index, HDL cholesterol, LDL cholesterol, triglyceride, γ-glutamyl transpeptidase, HbA1c, estimated glomerular filtration rate, smoking, physical activity, walking speed, sleeping, alcohol consumption, creatinine, sodium, potassium, salt intake, bone mineral density, body fat percentage, muscle mass, basal metabolic rate, and the number of teeth. All statistical analyses were performed using SPSS version 26 (IBM, Armonk, NY, USA). Two-tailed *p*-values were calculated for all analyses.

## 3. Results

### 3.1. Participant Characteristics

The participants’ characteristics are summarized in Table 1. Overall, 702 participants were enrolled in the study, and 220 were excluded due to missing data. Ultimately, 481 participants were included in the analysis. Of the participants, 223 (46.4%) were male, and the mean age was 66.7 (SD: 7.4) years. The mean body mass index was 66.7 (SD: 7.4) kg/m^2^. The mean HDL cholesterol level was 61.7 (SD: 15.1) mg/dL. The mean LDL cholesterol level was 121.8 (SD: 27.4) mg/dL. The mean triglyceride level was 101.9 (SD: 65.3) mg/dL. The mean gamma-glutamyl transpeptidase level was 37.7 (SD: 54.1) IU/L. The mean HbA1c was 6.0 (SD: 7.0). The mean estimated glomerular filtration rate was 69.4 (SD: 13.1) mL/min/1.73 m^2^. The mean creatinine was 85.9 (SD: 55.9) mL/min. The mean sodium was 119.7 (SD: 56.3) mEq/day. The mean potassium was 54.5 (SD: 30.7) mEq/day. The mean salt intake was 9.5 (SD: 2.1) g/day. The mean bone mineral density was 88.3 (SD: 12.2). The mean body fat percentage was 24.2% (SD: 8.9%). The mean muscle mass was 41.2% (SD: 8.5%). The mean basal metabolic rate was 1208.3 (SD: 230.8) kcal/day. The mean number of teeth was 23.5 (SD: 7.8). There were 41 (8.5%) smokers. From the questionnaire, 261 (54.3%) participants answered that they do physical exercises on a daily basis, 209 (43.5%) answered that their walking speed was fast, and 350 (72.8) respondents answered that they slept well. Eating speed was fast in 136 (28.3%) patients, normal in 301 (62.6%), and slow in 44 (9.1%). There were 133 (27.7%) participants that drank alcohol every day, 105 (21.8%) who did sometimes, and 243 (50.5%) who never did.

### 3.2. Univariate and Multivariate Logistic Regression Analysis

The results of univariate and multivariate logistic regression analysis are shown in Table 2. There were no significant associations in univariate analysis between eating speed and gender (odds ratio [95% confidence interval]: 1.00 [0.67–1.49]), age (0.99 [0.97–1.02]), HDL cholesterol (0.99 [0.97–1.00]), LDL cholesterol (1.00 [0.99–1.01]), triglyceride (1.00 [1.00–1.00]), gamma-glutamyl transpeptidase (1.00 [1.00–1.01]), estimated glomerular filtration rate (1.00 [0.98–1.01]), creatinine (1.00 [1.00–1.00]), sodium (1.00 [1.00–1.01]), potassium (1.00 [0.99–1.00]), bone mineral density (1.00 [0.99–1.02]), body fat percentage (1.01 [0.99–1.04]), muscle mass (1.02 [1.00–1.04]), basal metabolic rate (1.00 [1.00–1.00]), number of teeth (1.01 [0.98–1.04]), smoking (1.08 [0.53–2.23]), physical activity (0.95 [0.64–1.42]), walking speed (0.89 [0.59–1.32]), or drinking alcohol (1.15 [0.91–1.46]). However, univariate analysis revealed significant correlations between eating speed and body mass index (odds ratio [95% confidence interval]: 1.07 [1.02–1.13]), HbA1c (1.60 [1.18–2.18]), and salt intake (1.14 [1.04–1.26]). Because this study used the variable reduction method during logistic regression analysis, each variable was used once as an explanatory variable in the multiple analysis, but the following variables were finally extracted as strongly related items. Multivariate logistic regression analysis revealed significant correlations between eating speed and gender (2.15 [1.02–4.53]), HbA1c (1.60 [1.17–2.18]), salt intake (1.11 [1.01–1.22]), muscle mass (1.05 [1.00–1.09]), and sleeping (1.60 [1.03–2.50]).

## 4. Discussion

Our major findings in this study are that fast eating might lead to systemic diseases such as type 2 diabetes, renal dysfunction, and hypertension even after adjusting for the number of teeth. A 5-year cohort study analyzing diabetes incidence in 4853 Japanese participants reported a hazard rate of 2.08 for diabetes incidence at fast eating speeds compared with slow eating speeds [7]. In a study of Japanese adults that investigated the relationship between eating speed and poor glycemic control, several reports indicated that fast eating was associated with poor glycemic control, a measure of postprandial blood glucose [28,29]. In addition, fast eating speed may be independently associated with insulin resistance in the Japanese population [30]. Since the results of this study are consistent with previous reports, we believe that eating speed is also associated with glycemic control in healthy older people living in the area we studied. However, because many of the reported studies were based on data from Japan, the results of this study may differ depending on the effect of race. In fact, as a Japanese-specific report, impaired glucose tolerance, which results in postprandial hyperglycemia, is common among young, thin Japanese women with a BMI < 18.5 kg/m^2^, and is associated with insulin resistance and adipose tissue abnormalities as the cause of such impaired glucose tolerance [31]. Therefore, whether the results of this study can be applied to populations other than the Japanese population should be carefully determined. In basic research, histamine neurons have been reported to be involved in regulating masticatory function, particularly eating speed, in experiments using rats. Although histamine neurons in the brain are often explained as being activated by slower eating speeds to facilitate visceral fat burning via the sympathetic nervous system, the detailed mechanism is still unclear [32]. However, the strength of this study is that we reported results adjusted for masticatory function as a confounding factor. Although a relationship between masticatory function and eating speed has been reported, the masticatory function has not been considered a factor related to eating speed in many studies [19]. Therefore, it is important to note that the results of this study are based on an analysis that considers oral and dental health status. However, whether masticatory function positively or negatively correlates with eating speed is a controversial issue that must be considered.

A review of the literature was conducted on the relationship between eating speed and salt intake, but no data were obtained to show a relationship between the two. However, it has been reported that sensitivity to salt increases as the number of chews increases with the inoculation of hard foods [33]. Since mastication speed is generally considered to slow down as the number of chews increases, someone with a slower eating speed may have increased sensitivity to salt intake and may have suppressed salt intake beyond what is necessary [19]. A review has reported that lower salt intake is associated with a lower risk of cardiovascular disease, all-cause mortality, kidney disease, stomach cancer, and osteoporosis [34]. Therefore, while health guidance generally teaches the limitation of salt intake, correction of fast eating may be more effective as early preventive health guidance.

Multiple reports on gender differences in eating speed indicate that males are generally faster than females [35]. The presence of gender-related differences in eating speed is consistent with our findings, as eating speed is thought to be dependent on body size and bite size. Regarding muscle mass, slow eating speed has been reported to decrease muscle mass in people with type 2 diabetes [36]. In addition, slow eating has been reported to increase the likelihood of sarcopenia and undernutrition as well as loss of muscle mass, and the results of our study were similar [27,37]. However, more detailed cohort studies are needed to determine the causal relationship between muscle mass and eating speed, whether muscle mass is reduced because of slower eating speed, or whether the eating speed is reduced due to decreased function of masticatory muscle groups and muscle groups related to swallowing caused by reduced muscle mass. The relationship between sleep and eating speed was difficult to logically examine because no previous studies have pointed to a link between the two.

One of the concerns in this study was whether oral status was related to the speed of food intake. Whether oral status would increase or decrease the speed of food intake was also difficult to predict. This is because if the number of teeth is high, the person can bite well and thus may eat more slowly; on the other hand, the person may bite more efficiently and thus may eat more quickly. At least for these two conflicting hypotheses, our results suggest that the number of teeth is not related to the food intake speed and that other factors (e.g., swallowing function, cognitive function, etc.) may be involved as determining factors. The central pattern generator (CPG) in the medulla oblongata of the brainstem is believed to contribute to the formation of the motor patterns of masticatory swallowing movements [38]. Because the cerebral cortex is believed to trigger swallowing and regulate the sequential activity of the brainstem, aging may alter the speed of food intake by disrupting this regulation [39]. In fact, it has been reported that patients with amyotrophic lateral sclerosis, a neurological disease, have problems with food intake due to dysfunction of the CPG in dysphagia, which prevents smooth swallowing [40]. Therefore, based on the results of this study, we hypothesized that the factors that influence the speed of food intake can be divided into three categories: background factors (gender, age, and body size), oral status (masticatory function, swallowing function, tongue pressure, oral dryness, oral hygiene, and tongue and lip motor function), and neural functions connecting the brain and the oral cavity, which were not investigated in this study. Mouthful volume may also be a factor that influences the food intake speed as a separate factor from oral function. Past reports indicate that there is a negative correlation between mouthful volume and the number of bites and that the number of bites per mouthful volume decreased as the mouthful volume increased [41]. New studies are needed to investigate the determinants of food intake speed in more detail.

Several intervention studies have been conducted to control fast eating speed. The first was a randomized crossover design study in which participants were instructed to consume food under two dietary conditions, with slow eating resulting in significantly lower dietary energy intake [42]. The two conditions were a fast eating group (eat as quickly as possible: take large bites, chew quickly, and refrain from pausing and putting the spoon down between bites) and a slow eating group (eat as slowly as possible: take small bites, chew each bite thoroughly, and pause and put the spoon down between bites) [42]. The second study, which showed an association between fast eating speed and elevated blood glucose levels in a randomized crossover controlled trial on a Japanese female population, used an intervention method that set a time frame (fast group eating the test food in 10 min and slow group in 20 min) [13]. Another randomized controlled trial in a Japanese population also used a time frame intervention method (fast group in 5 min and slow group in 15 min) [43]. Considering these past studies, dental professionals should contribute to the prevention of systemic diseases by providing the following dietary guidance after restoring normal eating function by improving oral and swallowing conditions. The instructional approach to food intake should include chewing with small mouth openings, chewing thoroughly when chewing, and placing the spoon down on the table as often as possible to increase the spacing between food transports. It would be better to instruct them that each meal should take at least 20 min [13]. The key point here is the determination of oral and swallowing status. All previous studies are based on healthy individuals, albeit of different genders. In other words, the benefit of this slow eating is based on the assumption that there are no oral or systemic problems. It is important for dentists and dental hygienists to demonstrate their expertise in making this determination.

In the Japanese healthcare system, dental hygienists can provide practical guidance on oral and dental hygiene under the direction of dentists. In general, the main purpose of oral and dental hygiene instruction is to prevent dental diseases, and nutritional and dietary guidance are rarely provided so far [44]. Moreover, dental hygienists have rarely provided instruction or guidance that focuses on fast eating. However, considering the potential for widespread systemic effects of fast eating as suggested in the present study, it is necessary to provide guidance on eating too fast, as well as on dietary balance. Eating slowly, or chewing well not only increases saliva production and improves digestive efficiency, but also has social significance, such as the enjoyment of taste [45]. Therefore, such dietary and lifestyle guidance carried out by dental hygienists, along with the restoration of oral function by dentists, could be considered important in contributing to overall systemic healthcare and preventing systemic diseases such as type 2 diabetes, renal dysfunction, and hypertension. In Japan, the collaboration between medicine and dentistry has been strengthened since 2012, with dentists and dental hygienists playing an increasingly active role in both clinical practice and research [46]. It is hoped that studies with even larger sample sizes will become possible in the future and that the factors that make up fast eating speed and its effects on the whole body will be clarified.

This study has four main limitations. First, the lack of detailed data on medications and pre-existing treatment status in this study limits adjustment in multivariate analysis.

Second, the causal relationship between the relevant items is unknown because this was a cross-sectional study. Third, dietary speed was self-reported, which might lead to low objectivity and reliability. Finally, there was methodological bias such as the examinee and reporting bias. Future studies are needed to prove this causal relationship through a prospective cohort study that incorporates data on oral function.

## 5. Conclusions

Fast eating may be associated with overall health and lifestyle. The characteristics of fast eaters, after taking oral and dental status into consideration, tended to increase the risk of type 2 diabetes, renal dysfunction, and hypertension. Dental professionals should provide dietary and lifestyle guidance to fast eaters. Additionally, the number of teeth may not be associated with fast eating.

## Figures and Tables

**Table 1 healthcare-11-00646-t001:** Demographic data of participants (N = 481).

Variables	Item	n (%) or Mean ± SD
Gender	Male	223 (46.4)
	Female	258 (53.6)
Age (years)		66.7 ± 7.4
Body mass index (kg/m^2^)		23.0 ± 3.7
HDL cholesterol (mg/dL)		61.7 ± 15.1
LDL cholesterol (mg/dL)		121.8 ± 27.4
Triglyceride (mg/dL)		101.9 ± 65.3
Gamma-glutamyl transpeptidase (IU/L)		37.7 ± 54.1
Hemoglobin A1c (%)		6.0 ± 0.7
Estimated glomerular filtration rate (mL/min/1.73 m^2^)		69.4 ± 13.1
Creatinine (mL/min)		85.9 ± 55.9
Sodium (Na) (mEq/day)		119.7 ± 56.3
Potassium (K) (mEq/day)		54.5 ± 30.7
Salt intake (g/day)		9.5 ± 2.1
Bone mineral density (%YAM)		88.3 ± 12.2
Body fat percentage (%)		24.2 ± 8.9
Muscle mass (%)		41.2 ± 8.5
Basal metabolic rate (kcal/day)		1208.3 ± 230.8
Number of teeth		23.5 ± 7.8
Smoking	Yes	41 (8.5)
Physical activity	Yes	261 (54.3)
Walking speed	Yes	209 (43.5)
Food intake speed	early	136 (28.3)
	normal	301 (62.6)
	slow	44 (9.1)
Sleeping	Yes	350 (72.8)
Drinking alcohol	Every day	133 (27.7)
	Sometime	105 (21.8)
	None	243 (50.5)

**Table 2 healthcare-11-00646-t002:** Factors associated with food intake speed using univariate and multivariate logistic regression analysis (N = 481).

Variables	Univariate		Multivariate	
	Odds (95% CI)	*p*-Value	Odds (95% CI)	*p*-Value
Gender (Male)	1.00 (0.67–1.49)	0.99	2.15 (1.02–4.53)	0.05 *
Age	0.99 (0.97–1.02)	0.55		
Body mass index	1.07 (1.02–1.13)	0.01 *		
HDL cholesterol	0.99 (0.97–1.00)	0.04		
LDL cholesterol	1.00 (0.99–1.01)	0.54		
Triglyceride	1.00 (1.00–1.00)	0.66		
Gamma-glutamyl transpeptidase	1.00 (1.00–1.01)	0.33		
Hemoglobin A1c	1.60 (1.18–2.18)	<0.01 *	1.60 (1.17–2.18)	<0.01 *
Estimated glomerular filtration rate	1.00 (0.98–1.01)	0.54		
Creatinine	1.00 (1.00–1.00)	0.35		
Sodium (Na)	1.00 (1.00–1.01)	0.46		
Potassium (K)	1.00 (0.99–1.00)	0.21		
Salt intake	1.14 (1.04–1.26)	<0.01 *	1.11 (1.01–1.22)	0.04 *
Bone mineral density	1.00 (0.99–1.02)	0.90		
Body fat percentage	1.01 (0.99–1.04)	0.21		
Muscle mass	1.02 (1.00–1.04)	0.13	1.05 (1.00–1.09)	0.04 *
Basal metabolic rate	1.00 (1.00–1.00)	0.06		
Number of teeth	1.01 (0.98–1.04)	0.46		
Smoking (Yes)	1.08 (0.53–2.23)	0.83		
Physical activity (Yes)	0.95 (0.64–1.42)	0.81		
Walking speed (Yes)	0.89 (0.59–1.32)	0.55		
Sleeping (Yes)	1.56 (1.02–2.41)	0.04 *	1.60 (1.03–2.50)	0.04 *
Drinking alcohol (Every day)	1.15 (0.91–1.46)	0.24		

*: *p* < 0.05

## Data Availability

Not applicable.

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
