# Peer review of "Fast Eating Speed Could Be Associated with HbA1c and Salt Intake Even after Adjusting for Oral Health Status: A Cross-Sectional Study"

_healthcare, 2023, doi:10.3390/healthcare11050646_

Round 1

Reviewer 1 Report

Methodological Biases exist

Some References are missing

Author Response

Response to Comments/Suggestions

from Reviewer 1

Dear reviewer 1

We are truly grateful to your critical comments and thoughtful suggestions on our manuscript. Based on these comments and suggestions, we have made careful modifications into our original manuscript. All changes made to the main text are in red. We here would like to say thank you for your helpful suggestions, which were very supportive for further improving this manuscript. You may kindly find our point-by-point responses to your comments/ questions as below.

Sincerely yours

Professor. Takahiro Kanno, Corresponding Author for this article: healthcare- 2201030

Comments to the Author

Methodological Biases exist.

Response: Thank you very much for your comments. As you pointed out, we have added to the limitation that there is a methodological bias.

Some References are missing

Response: Thank you very much for your comments. As you pointed out, we have added references to all the areas you pointed out.

Where did the Authors use such variables? Did they use a standardized questionnaire? If so, state References...

Response: Thank you very much for your comments. This health examination included a standardized questionnaire, blood tests, urine tests, and an examination of the oral cavity. This detail was added by citing the literature.

How did the Authors determine the study size? Protocol? References?

Response: Thank you very much for your comments. Since this study used all data from previous health examinations, no sample size calculations were performed. The details of this have been added with additional references.

Responses to other comments

Response: Thank you very much for your comments. We have replaced all the words you pointed out as instructed.

We would also like to add that we have made additions to the discussion, etc., in response to points raised by the editorial board members.

Reviewer 2 Report

Reviewer’s results for the article of healthcare 220103

Major comments

 This is the article to study an association of eating rate with the other parameters among older adults of database obtained from health examinations. The authors have written the other two articles using the same subjects with different variables. The authors found as the result that the faster eating speed was associated with HbA1c in addition of male, more salt intake, higher body fat percentage, and less sleeping time shown in table 2. Especially faster eating speed and higher HbA1c might be the first to publish. My concerns are as the follows:

1.       How do we interpret this result? The person eating fast has higher HbA1c, and how many HbA1c higher does he or she have?

2.       Do participants have take anti-hyperglycemia drugs? In the other words, is there effects of these drug behind the results, or if the participants taking anti-hyperglycemic drug are omitted, the results of association might be stronger or different?

3.       How many participants who have anti-hyperglycemia among subjects? If very small, they might be omitted from participants?

As my result, I will review again after the authors answer to my abovementioned questions.

Minor comments

1.       Page8 line 328, the pages of reference 34 must be written.

2.       The authors used two words, eating speed and rate. What difference there are between two?

Author Response

Response to Comments/Suggestions

from Reviewer 2

Dear reviewer 2

We are truly grateful to your critical comments and thoughtful suggestions on our manuscript. Based on these comments and suggestions, we have made careful modifications into our original manuscript. All changes made to the main text are in red. We here would like to say thank you for your helpful suggestions, which were very supportive for further improving this manuscript. You may kindly find our point-by-point responses to your comments/ questions as below.

Sincerely yours

Professor. Takahiro Kanno, Corresponding Author for this article: healthcare- 2201030

Comments to the Author

Major comments

 This is the article to study an association of eating rate with the other parameters among older adults of database obtained from health examinations. The authors have written the other two articles using the same subjects with different variables. The authors found as the result that the faster eating speed was associated with HbA1c in addition of male, more salt intake, higher body fat percentage, and less sleeping time shown in table 2. Especially faster eating speed and higher HbA1c might be the first to publish. My concerns are as the follows:

How do we interpret this result? The person eating fast has higher HbA1c, and how many HbA1c higher does he or she have?

Response: Thank you very much for your comments. Although we cannot give specific HbA1c values, the results of this study can be interpreted to mean that people who eat quickly may have 1.6 times higher HbA1c values than those who do not.

Do participants have take anti-hyperglycemia drugs? In the other words, is there effects of these drug behind the results, or if the participants taking anti-hyperglycemic drug are omitted, the results of association might be stronger or different?

Response: Thank you very much for your comments. Since we do not have detailed information on medications, we have not adjusted the medication information. As you pointed out, it is quite possible that the underlying disease or its treatment status could have an impact, therefore we have added it to the limitation.

How many participants who have anti-hyperglycemia among subjects? If very small, they might be omitted from participants?As my result, I will review again after the authors answer to my abovementioned questions.

Response: Thank you very much for your comments. Since there is no detailed medication information as described above, this issue is a limitation. Therefore, this issue is also listed first in the limitation because it is a relatively important limitation in this study.

Minor comments

Page8 line 328, the pages of reference 34 must be written.

Response: Thank you very much for your comments. As you indicated, we have included the page numbers in reference no. 34.

The authors used two words, eating speed and rate. What difference there are between two?

Response: Thank you very much for your comments. There is no difference between the two terms. As you pointed out, we have unified everything to eating speed.

We would also like to add that we have made additions to the discussion, etc., in response to points raised by the editorial board members.

Round 2

Author Response

Response to Comments/Suggestions

from Reviewer 1

Dear reviewer 1

We are truly grateful to your critical comments and thoughtful suggestions on our manuscript. Based on these comments and suggestions, we have made careful modifications into our original manuscript. All changes made to the main text are in red. We here would like to say thank you for your helpful suggestions, which were very supportive for further improving this manuscript. You may kindly find our point-by-point responses to your comments/ questions as below.

Sincerely yours

Professor. Takahiro Kanno, Corresponding Author for this article: healthcare- 2201030

Comments to the Author

Please clarify the type of the article, eg. Research?

Response: Thank you very much for your comments. As you pointed out, we have clarify the type of the article.

Some References and words are missing

Response: Thank you very much for your comments. As you pointed out, we have added references to all the areas you pointed out.

How did the Authors determine the study size? Protocol? References?

Response: Thank you very much for your comments. Since this study used all data from previous health examinations, no sample size calculations were performed. The details of this have been added with additional references.